# KNOW THYSELF: TRANSFERABLE VISUAL CONTROL POLICIES THROUGH ROBOT-AWARENESS

**Edward S. Hu**     **Kun Huang**     **Oleh Rybkin**     **Dinesh Jayaraman**
GRASP Lab, University of Pennsylvania
`{hued, huangkun, oleh, dineshj}@seas.upenn.edu`

## ABSTRACT

Training visual control policies from scratch on a new robot typically requires generating large amounts of robot-specific data. How might we leverage data previously collected on another robot to reduce or even completely remove this need for robot-specific data? We propose a "robot-aware control" paradigm that achieves this by exploiting readily available knowledge about the robot. We then instantiate this in a robot-aware model-based RL policy by training modular dynamics models that couple a transferable, robot-aware world dynamics module with a robot-specific, potentially analytical, robot dynamics module. This also enables us to set up visual planning costs that separately consider the robot agent and the world. Our experiments on tabletop manipulation tasks with simulated and real robots demonstrate that these plug-in improvements dramatically boost the transferability of visual model-based RL policies, even permitting zero-shot transfer of visual manipulation skills onto new robots. Project website: `https://www.seas.upenn.edu/~hued/rac`

## 1 INTRODUCTION

Raw visual observations provide a versatile, high-bandwidth, and low-cost information stream for robot control policies. However, despite the huge strides in machine learning for computer vision tasks in the last decade, extracting actionable information from images remains challenging. As a result, even simple robotic tasks such as vision-based planar object pushing commonly require data collected over many hours of robot interaction to learn effective policies. This data collection cost would be amortized if the learned policies could transfer reliably and easily to new target robots. For example, a hospital adding a new robot to its robot fleet could simply plug in their existing policies and start using it immediately. Going further, other hospitals looking to automate the same tasks could purchase a robot of their choice and download the same policy models.

However, such transferable policies are difficult to achieve in practice. Even when the task setting, such as the hospital, remains unchanged, the changed visual appearance of the robot itself leads to out-of-distribution inputs for visual policies pre-trained on other robots. This issue particularly affects manipulation tasks: manipulation involves operating in intimate proximity with the environment, and any cameras set up to observe the environment cannot avoid also observing the robot.

There is a way out of this bind: most robots are capable of highly precise proprioception and kinesthesis to sense body poses and movements through internal sensors. We propose to develop "robot-aware" policies that can benefit from distinguishing pixels corresponding to the robot agent from those corresponding to the rest of the "world" in image observations.

Our robot-aware policies treat the robot and the world differently, to their advantage. While this general principle applies broadly to all visual controllers, we demonstrate the advantages of robot awareness in model-based reinforcement learning (MBRL) policies. MBRL policies work by planning through visual dynamics models that are trained to predict the consequences of agent actions. First, we inject robot awareness into the dynamics model, by factorizing it into robot-specific "robot" dynamics and robot-aware "world" dynamics. Figure 1 shows a schematic. Our experiments show that composing these two modules permits reliably transferring visual dynamics models even across robots that look and move very differently. Next, we design a robot-aware planning cost over the separated robot and world pixels. We show that this not only allows visual task reward specifications to transfer from a source to a target robot, it even leads to gains on the source robot itself by allowing the policy to reason separately about the robot and its environment.

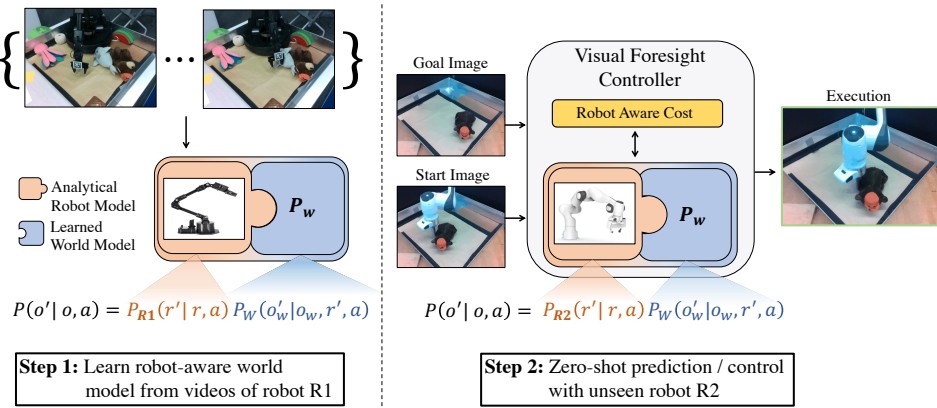

Figure 1: (Left) As part of our method, robot-aware control (RAC), we propose to factorize visual dynamics into a robot and world model. (Right) The world model is then readily reused with a new robot, permitting plug-and-play transfer.

We evaluate our method, robot-aware control (RAC), on simulated and real-world tabletop object pushing and pick-and-place tasks, demonstrating the importance of each robot-aware component. We show that a RAC policy trained on a single, low-cost 5-DoF WidowX arm can transfer entirely "zero-shot" to a very different 7-DoF Franka Panda arm in the real world. To our knowledge, this is the first demonstration of zero-shot visual skill transfer between real-world robots, and also the first instance of transfer after training on only one source robot.

## 2 Becoming Aware of the Robot Agent in the Scene

We set up our robot-aware control approach by starting with the question: what is required to be "aware of the robot" in a vision-based manipulation setting?

**First, a robot-aware agent must locate the robot in the visual observation from the camera.** Consider, for example, a manipulation task involving a standard robot arm, as in Fig 1. At each discrete timestep $t$, the input to the agent's policy is an RGB image observation $o(t)$ from a static camera observing the robot workspace. Since the robot must operate in close proximity with objects in the workspace, these images contain parts of the robot's body as well as the rest of the workspace. We define a visual projection function $I(\cdot)$ that maps from true robot state $r$ and "world" (i.e., the rest of the workspace) state $w$ to the image observation $o(t) = I(r(t), w(t))$. Robot states $r$ could be robot joint angles, and world states $w$ could be object poses in the workspace.

Note that the robot's state $r(t)$ is readily available in high precision through internal joint encoders. Further, a full geometric specification of the robot shape and joint degrees of freedom may be assumed available (usually from the manufacturer). Finally, techniques for camera-to-robot calibration are increasingly robust and efficient, to the point of acquiring calibration from even a single image when the robot shape is known as above (Labbé et al., 2021; Lee et al., 2020). Combining these, we may simply project the articulated robot shape through the camera parameters to obtain the visual projection of the robot in the scene, see Supp. A.11 for details. When parts of this projection may be occluded, we may further use RGB-D observations to easily identify and handle them. We represent this as an image segmentation mask $M_r(t)$ of the same height and width as image observations $o(t)$: $M_r$ is 1 on robot pixels and 0 elsewhere. We show examples of such masks in Fig 2. This simple process effectively spatially disentangles robot pixels from world pixels in image observations $o(t)$. This spatial disentanglement is a key building block for our robot-aware visual policies, since it permits treating robot and world pixels differently.[1]

Second, with common high-level robot action spaces used in reinforcement learning (such as end-effector displacement actions), **it may even be possible to analytically predict *future* robot states** $r'$ given the current state $r$ and the action $a$: $P_r(r, a) = r'$. This is not a strict requirement for robot-aware control, and we may instead simply learn the robot-specific dynamics model $P_r$ from a small amount of experience. However, as we will show in our experiments, analytical $P_r$ models work well for our manipulation tasks, and further, they permit fully zero-shot transfer to a new robot.

---

[1] Alternative techniques permit automatically "self-recognizing" the robot in a scene (Michel et al., 2004; Natale et al., 2007; Edsinger & Kemp, 2006; Yang et al., 2020) even without prior knowledge of robot embodiment.

## 3 TRANSFERABLE ROBOT-AWARE MODEL-BASED RL POLICIES

Above, we have described how a policy might be "robot-aware" by spatially disentangling the robot in its observations, and utilizing knowledge of its dynamics. We now show how such robot-awareness enables effective visual control policies that can be readily transferred between robots. In particular, we instantiate a robot-aware model-based reinforcement learning agent. Continuing in the manipulation setting from Sec 2, assume that tasks are specified through a goal image that exhibits the target configuration of the workspace. Given the goal image $o_g = I(r_g, w_g)$, the agent must execute a sequence of actions $\boldsymbol{a}_0^T = (a(0), a(t+1), ...a(T))$ to reach that goal, where $T$ is a time limit. Success is typically measured by how close the final world state is to the goal image. For example, the goal image might show a specific goal orientation of objects on a table, in which case success might be measured in terms of how close the objects are to their goals.

To solve such tasks, we inject robot-awareness into a widely used model-based reinforcement learning paradigm for robots: visual foresight (VF) (Finn & Levine, 2017; Ebert et al., 2018). Standard VF involves two key steps:

- **Visual dynamics modeling:** The first step is to perform exploratory data collection on the robot to generate a dataset $\mathcal{D}$ of transitions $(o(t), a(t), o(t+1))$. Dropping time indices, we sometimes use the shorthand $(o, a, o')$ to avoid clutter. Then, a visual dynamics model $P$ is trained on this dataset $\mathcal{D}$ to predict $o'$ given $o$ and $a$ as inputs, i.e. $o' \approx P(o, a)$. When the robot state $r$ is available, it is sometimes included as a third input to $P$ to assist in dynamics modeling.

- **Visual MPC:** Given the trained dynamics model $P$, VF approaches search over action sequences $\boldsymbol{a}_0^T$ to find the sequence whose outcome will be closest to the goal specification $o_g$, as predicted by $P$. For outcome prediction, they apply $P$ recursively, as $\hat{o}(o(0), \boldsymbol{a}) = P(P(\ldots P(o(0), a(0)), a(1)), \ldots a(T))$. Then, they pick the action sequence $\boldsymbol{a}^*(o(0), o_g) = \arg \min_{\boldsymbol{a}} C(\hat{o}(o(0), \boldsymbol{a}), o_g)$, where the cost function $C$ is commonly the mean pixel-wise $\ell_2$ error between the predicted image and the goal image. Sometimes, rather than measure the error of only the final image, the cost function may sum the errors of all intermediate predictions. For closed-loop control, only the first action $a^*(0)$ is executed; then, a new image $o(1)$ is observed, and a new optimal action sequence $\boldsymbol{a}_1^{T*}(o(1), o_g)$ is computed, and the process repeats. For action sequence optimization, we found the cross-entropy method (CEM) (De Boer et al., 2005) to be sufficient, although more sophisticated optimization methods (Zhang et al., 2019; Rybkin et al., 2021) could also work well.

To train reliable dynamics models $P$, visual foresight approaches commonly require many hours of exploratory robot interaction (Finn & Levine, 2017; Ebert et al., 2018) even for simple tasks. Data requirements may be reduced a little by interleaving model training with data collection: in this case, data is collected by selecting goal images and then running visual MPC using the most recently trained dynamics model to reach those goals. However, such goal selection must be designed to match the target task(s), which may trade off task generality for sample complexity.

### 3.1 IMPROVING VISUAL FORESIGHT THROUGH ROBOT-AWARENESS

Having trained one data-hungry visual dynamics model $P$ on one robot, do we still need to repeat this process from scratch on a new robot aiming to perform the same tasks? In standard VF, the answer is unfortunately yes, for two main reasons: (1) With the new robot, all observations $o(t) = I(r_{\text{new}}(t), w(t))$ are out-of-domain for $P$. Further, the visual dynamics of the new robot may be very different, as between a green 3-DOF robot arm, and a red 5-DOF robot arm. (2) As described above, standard VF approaches operate by aiming to match the goal image. However, since the observations contain the robot, they commonly require the task specification image $o_g = I(r_g, w_g)$ itself to contain the robot in a plausible goal-reaching pose. This makes task specification more difficult, and also robot-specific: it is impossible to plan to reach $o_g$ using a different robot. We show how VF policies may overcome these obstacles through robot-awareness.

#### 3.1.1 ROBOT-AWARE MODULAR VISUAL DYNAMICS

How can robot-awareness be useful in visual dynamics modelling? Recall from Sec 2 that the robot is both spatially disentangled and its dynamics are known to us as a function $P_r(r, a) = r'$, where $'$ denotes the next timestep. Here, we propose to exploit this by effectively factorizing the visual dynamics into world and robot dynamics terms.

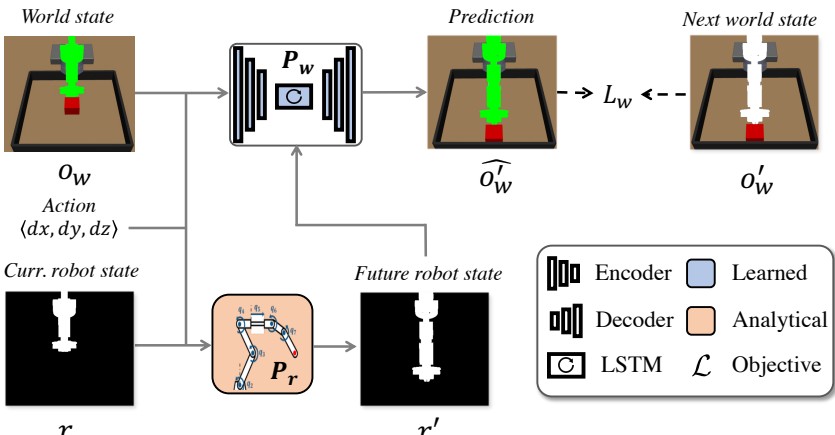

Figure 2: The visual dynamics architecture is composed of an analytical robot model $P_r$, and the learned $P_w$ world model. During robot transfer, the target robot's analytical model is used.

Given the robot state $r$ and the image observation $o$, we first compute the projected robot mask $M_r$ as above, then mask out the robot in the image to obtain $o_w = o \circ (1 - M_r)$, where $\circ$ is the pixel-wise product. See Fig 2 for examples of such masked images. Now, we train a world-only dynamics model $P_w$ by minimizing the following error, summed over all transitions in the training dataset $\mathcal{D}$:

$$\mathcal{L}_w = \|P_w(o_w, r, r', a) - o'_w\|, \tag{1}$$

where the next robot state is computed using the known robot dynamics, as $r' = P_r(r, a)$. Treating all dynamics models as probabilistic, this is equivalent to the following decomposition of full visual dynamics into a robot dynamics module $P_r$ and a world dynamics module $P_w$:

$$P(r', o'_w | r, o_w, a) = P_r(r' | r, a) P_w(o'_w | o_w, r, r', a). \tag{2}$$

**What advantage does this modularity offer?** First, since $P_w$ largely captures the physics of objects in the workspace, we hypothesize that it can be shared across very different robots. Second, $P_r$ is commonly available for every robot "out-of-the-box" as described in Sec 2, requiring no data collection or training. Together, this paves the way for zero-shot transfer of visual dynamics.

Specifically, suppose that a robot-aware world-only dynamics module $P_w$ has been trained on a robot arm $R_1$ with robot dynamics $P_{R1}$ for various manipulation tasks as seen in Figure 1. The full dynamics model, used during visual MPC, would be $P_1 = P_{R1}P_w$ as in equation 2. Then, given a new robot arm $R_2$ with dynamics $P_{R2}$, its full dynamics model $P_2 = P_{R2}P_w$ is available without any new data collection at all. We validate this in our experiments, demonstrating few-shot and zero-shot transfer of $P_w$ between very different robots.

### 3.1.2 ROBOT-AWARE PLANNING COSTS

Recall that the visual MPC stage in VF relies on the planning cost function $C(\hat{o}, o_g)$, which measures the distance between a predicted future observation $\hat{o}$ and the goal $o_g$. It is clear that for any task, the ideal planning cost is best specified as some function of the robot configurations $r$ and the world configurations $w$. However, since only image observations are available, it is common in VF to aim to minimize a pixel-wise error planning cost (Tian et al., 2019; Nair & Finn, 2020; Jayaraman et al., 2019), such as $C(\hat{o}, o_g) = \|\hat{o} - o_g\|_2^2$. Instead, our spatially disentangled observations $o \rightarrow (r, o_w)$ make it possible to produce the decomposed *robot-aware* cost:

$$C(\hat{r}, \hat{o}_w, r_g, o_{w,g}) = \lambda \|\hat{r} - r_g\| + \|\hat{o}_w - o_{w,g}\| \tag{3}$$

The first term in this expression is the robot cost scaled by $\lambda$ and the second term is the world cost. We compute a robot cost that is usable by all robots such as end effector distance, and compute the world cost by measuring pixel-wise distance over the world region of the predicted and goal image.

**Why should this decomposed cost help?** This robot-aware cost makes it possible to separately modulate the extent to which robot and world configurations affect the planning cost. To motivate this, observe that the basic pixel-wise cost suffers from a key problem: it is affected inordinately by the spatial extents of objects in the scene, so that large objects get weighted more than small objects. Indeed, Ebert et al. (2018) report that the robot arm itself frequently dominates the pixel cost in

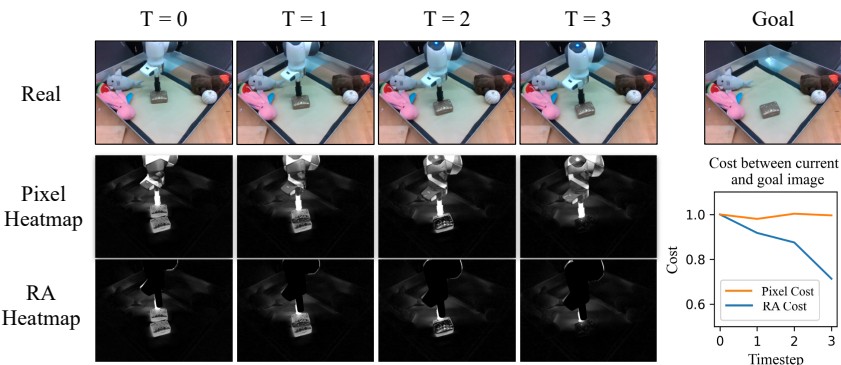

Figure 3: The behavior of the pixel and RA cost between the current images of a feasible trajectory and the goal image. (Left) The first row shows the trajectory. The next two rows show the heatmaps (pixel-wise norm) between the image and goal for each cost. The heatmaps show high cost values in the robot region, while the RA cost heatmaps correctly have zero cost in the robot region. (Right) The relative pixel cost fails to decrease as the trajectory progresses, while the RA cost correctly decreases.

manipulation. This means that the planner often selects actions that match the *robot position* in a goal image, ignoring the target objects. See Figure 3 for a visual example of the pixel cost behavior and its failure to compute meaningful costs for planning. As a result, even if the task involves displacing an object, the goal image is usually required to contain the robot in a plausible pose while completing the task (Nair & Finn, 2020). For our goal of robot transfer, this is an important obstacle, since the task specification is itself robot-specific. Even ignoring transfer, robot-dominant planning costs hurt performance, and gathering goal images with the robot is cumbersome. Our approach does not require robots to be in plausible task completion positions in the goal images. In fact, they may even be completely absent from the scene. In our experiments, we use goal images without robots, and sometimes with humans in place of robots. To handle them, we could set $\lambda$ to 0 in the above equation to instantiate a cost function that only focuses on the world region.

Some VF-based methods do handle goal images without robots; however, they typically require all input images for dynamics and planning to also omit the robot, so that image costs are exclusively influenced by the world configurations $w$ (Wang et al., 2019; Pathak et al., 2018; Agrawal et al., 2016). For closed-loop controllers, this often means extremely slow execution times, because, *at every timestep*, the robot must enter the scene, execute an action, and then move out of the camera view. This also eliminates tasks requiring dynamic motions.

Finally, there are efforts to learn more sophisticated cost functions over input images $C_\theta(\hat{o}, o_g)$ (Nair et al.; Sermanet et al., 2018; Srinivas et al., 2018; Yu et al., 2019; Tian et al., 2021). For example, Nair et al. train a latent representation to focus on portions of the image that are different between the goal and the current image, and show that costs computed over these latents permit better control on one robot. These approaches all *learn* the cost contributions of different objects or regions, from data. Instead, we directly segment the robot using readily available information. While we restrict our evaluation to basic pixel-based costs, we expect that these other costs will also benefit from spatially disentangled inputs, i.e., $C_\theta(\hat{r}, \hat{o}_w, r_g, o_{w,g})$.

## 3.2 IMPLEMENTATION DETAILS

We summarize some key implementation details about the robot-aware model here, and refer the rest to Supp. A.1. For implementing the learned world dynamics model $P_w$, we extended the authors' implementation of the SVG architecture (Denton & Fergus, 2018) to be conditioned on robot actions and states. SVG consists of a convolutional encoder, frame predictor LSTM, and decoder alongside a learned prior and posterior network. The encoder network takes in the RGB image, current mask, and future mask as a 5-channel 64x48 image. It outputs a spatial latent of dimension (256, 8, 6). Actions and end effector poses are then tiled onto this spatial latent before being fed into the convolutional LSTM. The convolutional LSTM output is then fed into the decoder which outputs the predicted image. Refer to algorithm 1 and 2 for world model training and testing pseudocode.

The world model $P_w$ is a CNN that must take the world pixels $o_w$ as input and predict future world pixels $o'_w$. Since it cannot selectively process partial images $o_w$, we must instead feed in a

full rectangular image. However, if we trained $P_w$ on full images of the training robots $o$ without modification, then test images with new, different-looking robots would lie outside the training distribution of $P_w$. To induce some invariance to the robot appearance in $P_w$, the robot is masked (producing the appearance of a black robot) at training as well as at test time. Note that the world model is still trained based on Eq. 1, which only penalizes errors in world pixels. We use MoveIt, PyRobot, and MuJoCo to implement the analytical robot dynamics for the various robots.

## 4 EXPERIMENTS

Our experiments focus on zero and few-shot transfer across robots for pushing and pick-and-place (Finn & Levine, 2017; Ebert et al., 2018; Dasari et al., 2019): a robot arm must move objects to target configurations, specified by a goal image, on a tabletop. We aim to answer: **(1)** How does the robot-aware controller compare against standard controllers and a domain adaptation baseline when transferred to a new robot? **(2)** Which robot-aware components in the visual foresight pipeline are most important for transfer?

**Baselines.** For evaluating robot-aware dynamics models, we compare against prior visual foresight methods (Ebert et al., 2018; Finn & Levine, 2017) which use action-conditioned models that solely rely on image input (VF), as well as models that rely on robot state and images (VF+State). For our full robot-aware pipeline, we experiment with combinations of predictive models and cost functions to probe their transferability to new robots. Hereafter, we will refer to all controllers by the names of the dynamics model and the cost, e.g., RA/RA for the full robot-aware controller, and VF+State/Pixel for the baseline VF controller. See Table 2 for all controllers. We also compare against an unsupervised domain translation approach (CycleGAN+VF+State/Pixel), based on the human-to-robot imitation policy of Smith et al. (2020), that learns a mapping between target and source robot images for transfer. Note that this policy is few-shot since it is trained on test robot images. See Supp. for additional details about baselines, calibration, data, training, and evaluation.

**Transfer settings.** We evaluate the transferability of robot-aware control in various configurations:

▷ Simulated zero-shot transfer from one robot. Simulation permits a "straight swap" of the robot with perfectly controlled viewpoints and environments. We conduct pushing and pick-and-place experiments by training policies on a 5-DoF WidowX200 arm and evaluating these models zero-shot on an unseen 7-DoF Fetch arm. For the prediction experiment, we train on 10k trajectories of length 30 of the WidowX200 arm performing random actions on the tabletop with several objects, and evaluate on 1000 trajectories of the Fetch robot. See Supp. A.9 for more prediction experiments. For the control experiments, we transfer the WidowX200 trained policies to the Fetch robot and perform pushing and pick-and-place tasks. The CycleGAN (Zhu et al., 2017) (authors' implementation) was trained using 1k videos (12k frames) each of the training and test time robots. See Supp. A.3 for more details on control experiments.

▷ Real zero-shot transfer from one / multiple robot(s). We evaluate zero-shot prediction and control of an unseen Franka robot (Fig. 4) from models trained on ∼1.8k videos of a single WidowX200 robot, as well as models trained on a multi-robot dataset (Fig. 4) that randomly perturbs the objects. The multi-robot models are pretrained on 83k videos including 82k RoboNet (Dasari et al., 2019) videos of Sawyer, Baxter, and WidowX robots, and 1k videos from the above WidowX200$^2$ dataset. We used a subset of RoboNet for which we were able to manually annotate camera calibration matrices and robot CAD models (see Supp. A.11). After training, we deploy the models to the unseen Franka and perform pushing tasks.

Next, we evaluate zero-shot predic­tion of an unseen Modified Wid­owX200 (Fig. 4) from models trained on the multi-robot dataset. To mod­ify the appearance and dynamics of the WidowX200, we swapped out the

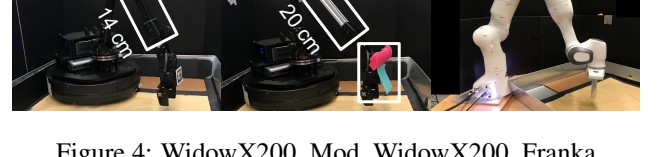

Figure 4: WidowX200, Mod. WidowX200, Franka

black 14cm forearm link with a silver 20cm forearm link, and added a foam bumper and sticker. We collected 25 videos of the modified robot for evaluation.

▷ Few-shot transfer. Finally, we follow the few-shot transfer setup of Robonet (Dasari et al., 2019) by first pretraining the models on RoboNet data, and then finetuning on ∼400 videos of the WidowX200. We then evaluate the prediction and control performance of the WidowX200 on pushing tasks.

---

$^2$Note that the WidowX and WidowX200 are different robots.

Table 1: World prediction evaluations, reported as average world region PSNR and SSIM over 5 timesteps. We compare against conventional models that use image and/or state (VF and VF+State).

| Model | Zero-shot Sim. Fetch (Train on WidowX200) | | Few-shot WidowX200 (Train on 3 robots) | | Zero-shot Mod. WidowX200 (Train on 4 robots) | | Zero-shot Franka (Train on 4 robots) | | Zero-shot Franka (Train on WidowX200) | |
|---|---|---|---|---|---|---|---|---|---|---|
| | PSNR ↑ | SSIM ↑ | PSNR ↑ | SSIM ↑ | PSNR ↑ | SSIM ↑ | PSNR ↑ | SSIM ↑ | PSNR ↑ | SSIM ↑ |
| VF | 39.9 | 0.985 | 32.86 | 0.939 | $30.59 \pm 3.18$ | $0.913 \pm 0.04$ | $28.31 \pm 3.35$ | $0.901 \pm 0.04$ | $28.42 \pm 3.43$ | $0.905 \pm 0.04$ |
| VF+State | - | - | 33.05 | 0.941 | $31.03 \pm 3.17$ | $0.915 \pm 0.04$ | $29.19 \pm 3.02$ | $0.901 \pm 0.04$ | $28.31 \pm 3.18$ | $0.894 \pm 0.05$ |
| RA (Ours) | **41.8** | **0.990** | **33.26** | **0.944** | **31.83** $\pm$ 2.88 | **0.924** $\pm$ 0.03 | **29.37** $\pm$ 2.93 | **0.904** $\pm$ 0.04 | **29.63** $\pm$ 3.00 | **0.928** $\pm$ 0.03 |

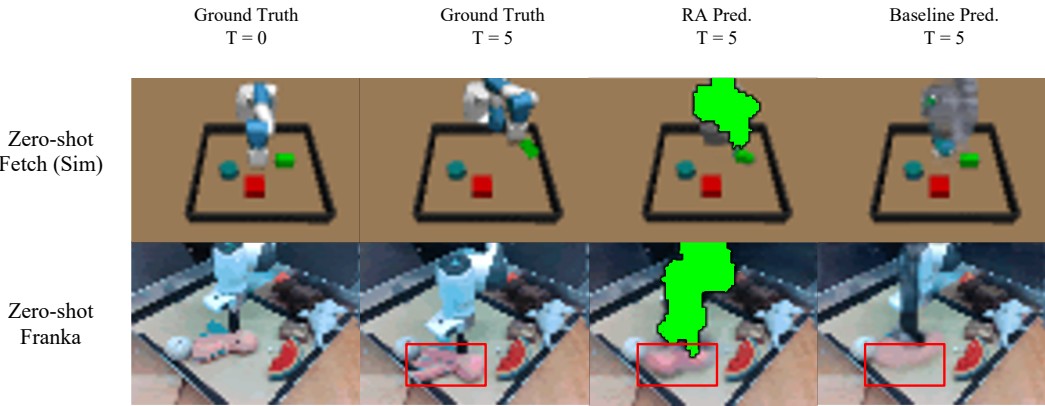

Figure 5: Model outputs of the zero-shot experiments. First row: the RA model is able to predict contact with the green box, while the baseline (VF+State) predicts none. Next row: the RA-model models the downward motion of the octopus more accurately. Refer to the website for videos.

**Results: Transferring visual dynamics models across robots.** the Table 1 shows quantitative prediction results for average PSNR and SSIM per timestep over a 5-step horizon, computed over the world region. Peak signal to noise ratio (PSNR) and structural similarity index metric (SSIM) are widely used metrics used for evaluating video prediction approaches. PSNR scales the inverse MSE error logarithmically, and SSIM measures similarity over local patches instead of per-pixel. The robot-aware world dynamics module $P_w$ outperforms the baselines across all zero-shot and few-shot transfer settings, simulated and real, on both metrics. To supplement the quantitative metrics, Fig 5 shows illustrative examples of 5-step prediction, where we recursively run visual dynamics models to predict images 5 steps out from an input.

We show more examples of prediction results in the Supp. and website. Some key observations: first, the VF+State baseline overlays the original robot over the target robot rather than moving the target robot (see Figure 5, bottom right). Next, RA produces relatively sharper object images and more accurate object motions, as with the pink octopus toy in Figure 5. Finally, while both VF+State and RA predict motions better for large objects, VF+State more frequently predicts too small, or even zero object motions (see Figure 5, top right). Our RA approach can use known robot dynamics for accurate robot state predictions, which in turn reduces uncertainty in predicting object interaction. As we will show next, these performance differences in object dynamics prediction are crucial for successfully executing manipulation tasks.

**Results: Transferring visual policies across robots.** We now evaluate our full robot-aware MBRL pipeline on simulated and real pushing tasks, as well as a simulated pick-and-place task. This jointly evaluates the RA dynamics models and the RA planning cost.[3] Pushing tasks are specified by a single goal image, while pick-and-place tasks are specified by three goals for picking, lifting, and placing. After each evaluation episode, we measure the distance between the moved object's current and goal position. We define a successful episode as a distance under 5cm for the pushing task, and under 2.5cm for the pick-and-place task. For each task, we use an object-only goal image where the

---

[3]We evaluate the RA cost in isolation by planning with perfect dynamics models in Supp. A.6. In Supp. A.7, we evaluate the RA cost with human goal images.

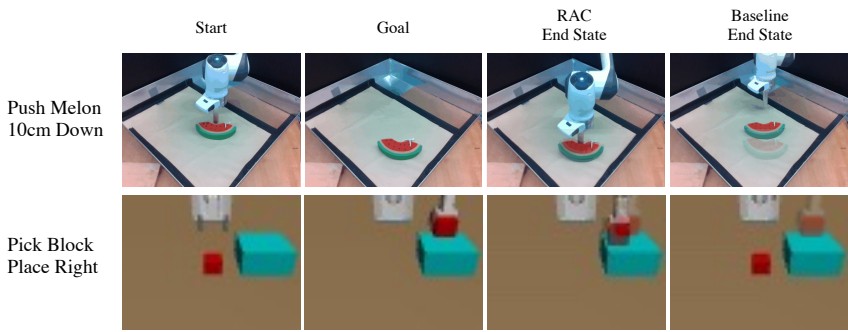

Figure 6: Example results from the control experiments (full videos on website). We show the start, goal, and end states of RA/RA and baseline (VF+State/Pixel). The goal image is overlaid on top of the end state for visual reference.

Table 2: Zero-shot control results. All models are trained on a single robot (WidowX200) unless denoted otherwise. Note the CycleGAN baseline is trained on 12k images of the test-time robot.

| Dynamics Model | Cost | Fetch Push (Sim.) | Fetch Pick-and-place (Sim.) | Franka Push (Real) |
|---|---|---|---|---|
| CycleGAN+VF+State | Pixel | 12/20 (60%) | 0/20 (0%) | - |
| VF+State | Pixel | 0/20 (0%) | 0/20 (0%) | 0/30 (0%) |
| RA | Pixel | 0/20 (0%) | 0/20 (0%) | 0/30 (0%) |
| VF+State | RA | 12/20 (60%) | 4/20 (20%) | 6/30 (20%) |
| RA | RA | **18/20** (90%) | **8/20** (40%) | **22/30** (71%) |
| VF+State (Multi-robot) | Pixel | - | - | 11/30 (36%) |
| RA (Multi-robot) | RA | - | - | **27/30** (90%) |

object is in the correct configuration, but the robot is not in the image. For pushing, we vary the goal position and object per trial and for pick-and-place, we vary the block initialization and goal position. See Supp. A.3 and A.5 for more details.

Table 2 and 3 shows that RA/RA achieves the highest success rate in all tasks and transfer settings. Note that our method's success rate (40%) on pick-and-place is not much worse than our method's success rate of 55% on the source robot (see Supp. A.10 for all source robot results), which shows our method is robust to robot transfer. We observed two primary failure modes of the VF baselines in zero-shot scenarios. First, both VF+State/Pixel and RA/Pixel, which optimized the pixel cost, tend to retract the arm back into its base, because the cost penalizes the arm for contrasting against the background of the goal image as seen in Figure 6. Next, consistent with the prediction results, VF+State/Pixel and VF+State/RA suffer from predicting blurry images of the training robot and inaccurate object dynamics, which negatively impacts planning.

The CycleGAN baseline performs moderately well in pushing (60% success) but completely fails in pick-and-place (0% success). Qualitatively, the CycleGAN baseline correctly moves to pick the block, but nearly always fails by selecting an unstable grasp, where the block slips through the gripper. Note that the CycleGAN itself is trained correctly: it successfully produces visually high quality domain-translated images of Fetch-to-WidowX (see website). However, transfering dynamics models for grasping requires very precise domain translation, which CycleGAN is unable to achieve even when trained with 12k images of the target robot.

The failure of the baselines to transfer zero-shot is expected, since generalizing to a new robot from training on a single robot dataset is a daunting task. One natural improvement is to train on multiple robots before transfer to facilitate generalization. We evaluate RA/RA and VF+State/Pixel in this multi-robot pretraining setting, where they are trained on RoboNet videos in addition to our WidowX200 dataset. As Table 2 ("Multi-robot" rows) shows, the VF+State/Pixel improves with training on additional robots, but still falls far short of RA/RA trained on even a single robot. RA/RA improves still further with multi-robot pretraining.

▷ **Few-shot transfer.** In this setting, dynamics models are pretrained on RoboNet and finetuned on ∼400 videos of the WidowX200. As Table 3 shows, RA/RA significantly outperforms the baseline controller. Due to the baseline's inability to model object dynamics aside from the largest object, it succeeds only on pushes with the largest object. The RA model adequately models object dynamics for all object types, and is able to succeed at least once for each object (see Supp. A.5 for object details).

Table 3: Few-shot transfer to WidowX200 control results.

| Dynamics Model | Cost | Success |
|---|---|---|
| VF | Pixel | 4/50 (8%) |
| RA | RA | **40/50** (80%) |

## 5 OTHER RELEVANT PRIOR WORK

Several works have focused on transferring controllers between tasks (Duan et al., 2017; Finn et al., 2017b), environments (Finn et al., 2017a), and simulation-to-real (Tobin et al., 2017), but relatively little attention has been paid to few / zero-shot transfer across distinct robots. Aside from RoboNet (Dasari et al., 2019), discussed above, a few prior works have studied providing the robot morphology as input to model-free RL policies, enabling transfer to new robots (Chen et al., 2018; Devin et al., 2017; Wang et al., 2018).

Devin et al. (2017) train modular policies containing a robot module and a task module, both learned from data, but requires data gathered on the new robot for training the robot-specific module before transfer is possible. Chen et al. (2018) propose to train a "universal" policy conditioned on the robot hardware specification, but must train on many robots to permit transfer, and operates on low-dimensional states. In contrast, RAC transfers *visual* policies after training even on a *single* robot. For simulated snake and centipede robots with repeated chained segments, NerveNet (Wang et al., 2018) demonstrates transferable state-based locomotion skills by running a graph neural network policy over the known chain structure. This permits transfer, for example, from a centipede with 4 chained segments to one with 10 identical chained segments. RAC targets more challenging settings: visual object interaction and manipulation on real robots with varied morphologies and appearances. RAC is also technically distinct: it trains a model-based goal-reaching controller, while these prior methods (Devin et al., 2017; Chen et al., 2018; Wang et al., 2018) train model-free task-specific policies. Richard et al. (2021) concurrently propose a decomposed world and robot state dynamics model to enable zero-shot transfer across robots in state-based navigation with sim-to-real training, whereas we evaluate our method on transfer of vision-based manipulation skills and train on real data.

Other works train visual representations that generalize to different embodiments, such as humans or other robots, and use them to infer rewards for RL (Zakka et al., 2021; Zhou et al., 2021; Smith et al., 2020; Sermanet et al., 2018). However, these methods require access to images or significant experience on the test-time robot. For example, Zakka et al. (2021) learn task-specific rewards (e.g. block pushing) and require ∼100k steps on the test robot to learn. In contrast, RAC's world dynamics model can be reused across tasks, and can achieve multiple tasks through goal image specification.

Finally, our work has interesting connections to "contingency awareness" in cognitive science (Watson, 1966), that has also been studied for RL (Bellemare et al., 2021; Choi et al., 2018): in simple 2D Atari games, they show that learning to localize the agents in image observations can improve model-free visual reinforcement learning and exploration. We operate in more complex real-world robotic settings, and focus on transferring model-based policies across agents.

## 6 CONCLUSION

We have studied the challenging task of transferring learned visual control policies for object manipulation across robot arms that might be very different in their appearance and capabilities. Our "robot-aware control" (RAC) paradigm, when used in widely used model-based RL algorithms, convincingly transfers *zero-shot* to unseen real and simulated robots for object manipulation for the first time, and also yields large gains for few-shot transfer. RAC benefits from a world model that aims to capture the physics of the environment independent of the robot so as to be fully transferable. Implemented with pixel models and costs, RAC's world model is still subject to limitations: it is tied to the end-effector action space and even to the shape of the source robots, which may inhibit transfer in more extreme settings. Further, the operating environment and its physics might differ from one robot to the next. We will aim to address these limitations and challenges in future work.

## 7 ACKNOWLEDGEMENTS AND DISCLOSURE OF FUNDING

This work was partly supported by an Amazon Research Award to DJ. The authors would like to thank Karl Schmeckpeper and Leon Kim for technical guidance, the anonymous reviewers for their constructive feedback, and the Perception, Action, and Learning Group (PAL) for general support.

## 8 REPRODUCIBILITY STATEMENT

To ensure reproducibility, we will release the codebase that contains our video prediction and control algorithms, as well as weights for our trained models. The supplementary contains details about hyperparameters (Supp. A.2) and architecture (Supp. A.1). For datasets, we will release our subset of RoboNet labeled with robotic masks, and our WidowX video dataset. We provide details of our mask labeling process in Supp. A.11, and the labeling code will be in the codebase. See the website for the code `https://www.seas.upenn.edu/~hued/rac`.

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

# A    APPENDIX

We encourage the reader to visit the website at the url: `https://www.seas.upenn.edu/~hued/rac` for an explanation video and additional video visualizations.

## A.1    NETWORK DETAILS

The robot-aware dynamics model described in Section 3.1.1 consists of two modules, an analytical robot-specific dynamics model $P_r$ and a learned, visual dynamics world model $P_w$. The robot dynamics model is required to predict the next robot state $r'$ (end effector pose and mask) from the current robot state $r$ and action $a$ and feed that as input into the world model. This future robot state and mask is useful for the world model to predict the overall future state, as it can infer object displacement from such information.

**Neural network architecture for the learned visual dynamics $P_w$.** We extend the stochastic video generation (SVG) architecture (Denton & Fergus, 2018) which consists of the convolutional encoder, frame predictor LSTM, and decoder alongside a learned prior and posterior.

The encoder consists of 4 layers of VGG blocks (convolution, batch norm, leaky RELU) and passes skip connections to the decoder. We use a convolutional LSTM for the learned prior, posterior, and frame predictor. As seen in Figure 2, visual data such as the current RGB observation $o_w$ with dimension (3,48,64), current mask with dimension (1,48,64), and future mask $M_r, M_{r'}$ (1, 48, 64) are concatenated channel-wise and fed into the encoder, which convolves the input into a feature map of dimension (256, 8, 8). Additional data such as the action, current state, and future state are tiled onto the feature map before being passed into the recurrent frame predictor, outputting a feature map of size (256, 8, 8) that gets decoded into the next RGB image $o'$.

**Generating predictions from the dynamics model at test time.** During training, the RA-model utilizes ground truth future robot state as input for prediction since all data is recorded in advance. From our prediction results in Table 1, we find that using future robot state and masks to be useful for future prediction. But how might we produce the future input $r'$ for $P_w$ during test time?

We use the analytical dynamics model $P_r$ as described in Section 3.1.1 to compute the future inputs. We now describe this process with more detail. The analytical dynamics model $P_r$ predicts the future robot state $r'$ as a function of the current state $r$ and action $a$. For example, if the control commands are robot end-effector displacements, then the robot state $r$ is the gripper pose, and next state $r' = P(r, a) = r + a$. Then, to obtain the masks, we must compute the full robot pose and its visual projection in two simple steps. First, we compute the joint positions using inverse kinematics $q' = \text{IK}(r')$, and then project the joint positions into a mask $M_{r'} = \text{Project}(q')$. We use the analytical inverse kinematics solver from the PyRobot library to obtain future joint positions for the WidowX200 while for the Franka, we use the MoveIt! motion planning ROS package.

Next, we use a simulator such as MuJoCo that supports 3D rendering of the robot along with camera calibration to project the image of the virtual robot into a 2D mask. These steps are also important for recursively applying the dynamics models to produce "rollouts". Given a starting image $o(t)$ and actions $a(t+1), a(t+2), ..., a(T)$, we can predict $\hat{o}(t+1)$ as above. To predict $\hat{o}(t+2)$, $P_w$ needs as input $M(t+1), M(t+2), r(t+1), r(t+2)$, all of which are inferred as above through the analytical dynamics $P_r$. Algorithm 2 provides pseudocode to generate such rollouts.

**RA-loss implementation.** In Algorithm 3, we provide a PyTorch implementation of the robot-aware dynamics loss (eq. 1) over RGB images. Note that the principle of separating the robot region and world region in the loss computation is general, and can be applied to other state formats as well.

## A.2    VISUAL DYNAMICS EXPERIMENT DETAILS

We now present some details of the visual dynamics experiments in Section 1 where we evaluated models on unseen robots as seen in Figure 4.

**Training and finetuning details.** Models were pretrained for 150,000 gradient steps on the RoboNet dataset with the Adam optimizer, learning rate $3e-4$ and batch size of 16. We use scheduled sampling to change the prediction loss from 1-step to 5-step future prediction over the course of training. For

---

**Algorithm 1** TRAIN

---

1: **Input:** World model $W$, analytical model $R$, image sequence $o(1)\ldots o(T+1)$, mask sequence $M_r(1)\ldots M_r(T+1)$, state sequence $r(1)\ldots r(T+1)$, actions $a(1),...,a(T)$.
2: $\hat{O}_w \leftarrow \{\}$
3: $O_w \leftarrow \{(1-M_r(2))\circ o(2),\ldots,(1-M_r(T+1))\circ o(T+1)\}$
4: $o_w \leftarrow \text{MaskRobot}(o(1),M_r(1))$     // Here we abuse the $o_w$ notation to be the masked image
5: **for** $t=1$ **to** $T$ **do**
6:    $\hat{o}' \leftarrow W(o_w,r(t),r(t+1),a(t))$
7:    $\hat{o}_w(t+1) \leftarrow (1-M_r(t+1))\circ \hat{o}'$
8:    $\hat{O}_w \leftarrow \hat{O}_w \cup \hat{o}_w(t+1)$
9:    $o_w \leftarrow \text{MaskRobot}(\hat{o}',M_r(t+1))$
10: **end for**
11: Compute world loss (eq. 1) between $O_w$ and $\hat{O}_w$.
12:
13: **return**

---

**Algorithm 2** TEST

---

1: **Input:** World model $W$, analytical model $R$, Start image $o(1)$, start mask $M_r(1)$, start state $r(1)$, actions $a(1),...,a(T)$.
2: $\hat{O}_w = \{\}$
3: $r \leftarrow r(1)$
4: $o_w \leftarrow \text{MaskRobot}(o(1),M_r(1))$     // Here we abuse the $o_w$ notation to be the masked image
5: **for** $t=1$ **to** $T$ **do**
6:    $r' \leftarrow R(r,a)$
7:    $\hat{o}' \leftarrow W(o_w,r,r',a(t))$
8:    $\hat{o}_w(t+1) \leftarrow (1-M_{r'})\circ \hat{o}'$
9:    $\hat{O}_w \leftarrow \hat{O}_w \cup \hat{o}_w(t+1)$
10:    $o_w \leftarrow \text{MaskRobot}(\hat{o}',M_{r'})$
11:    $r \leftarrow r'$
12: **end for**
13:
14: **return** $\hat{O}_w$

---

fine-tuning, all models were trained on the fine-tune dataset for 10,000 gradient steps with learning rate of $1e-4$, batch size of 10, and scheduled sampling. See algorithm 1 for pseudocode.

**Evaluation details.** To evaluate PSNR and SSIM metrics over the world region of the images instead of the entire image, we preprocess the images by setting the robot region of the images to black. This corresponds to removing all pixel differences between the predicted and target robot region before computing the PSNR and SSIM over the entire region.

For all experiments, we evaluate on sequences of length 5 and calculate the average world PSNR and SSIM metrics over the timestep. Because the models are stochastic, we perform Best-of-3 evaluation where 3 videos are sampled from the model, and we report the sample with the best PSNR and SSIM.

**Dataset.** Following RoboNet (Dasari et al., 2019) conventions, the image dimension is 64 by 48 pixels. and the action space of the robot is the displacement in end effector pose $(x,y,z,\theta,f)$ where $x,y,z$ are the cartesian coordinates of the gripper with respect to robot base, $\theta$ is the yaw of the gripper, and $f$ is the gripper force. The videos are length 31, but we sample subsequences of length 6 from the video to train the network to predict 5 images from 1 conditioning image. Due to the workspace bounds varying in size across robots, RoboNet chooses to normalize the cartesian coordinates to [0, 1] using the minimum and maximum boundary of each workspace.

The WidowX200 dataset consists of 1800 videos collected using a random gaussian action policy that outputs end effector displacements in the action space. For the pushing task, we fix $z,\theta,f$ to constants, and sample $x,y$ displacements from a 2 dimensional gaussian with 0 mean and $3.5\sigma$ in

**Algorithm 3** Pytorch code for the $L_w$ loss.

```
def ra_l1_loss(pred, target, mask):
    """
    pred and target are of images of dim (3, H, W)
    mask is binary segmentation mask of the robot of dim (1, H, W)
    outputs the mean L1 norm of world pixels
    """
    difference = target - pred # (3, H, W)
    # repeat mask across channel axis so we can use it
    # to index the robot region in difference
    robot_region = mask.repeat(3,1,1)
    # ignore the pixel differences in the robot region
    difference[robot_region] = 0
    # compute the l1 norm over the world pixels
    num_world_pixels = (~robot_region).sum()
    l1_per_pixel = diff.abs().sum() / num_world_pixels
    return l1_per_pixel
```

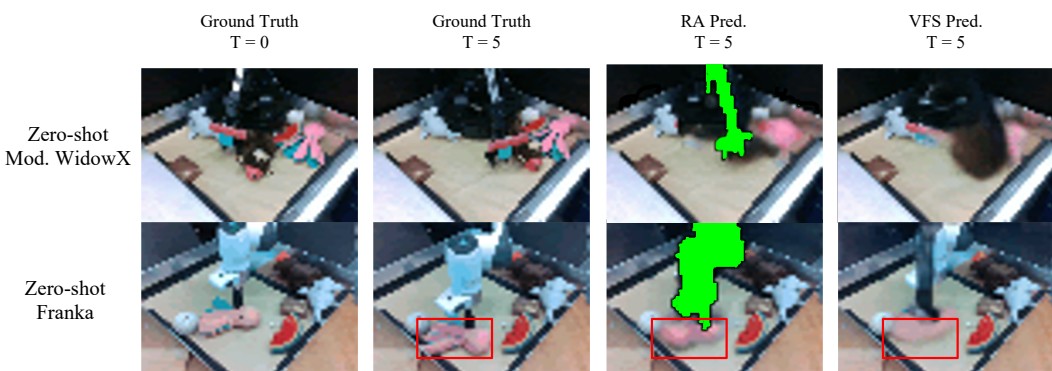

Figure 7: Qualitative outputs of the zero-shot prediction experiments on models that were pretrained on multiple robots. From left to right, we show the ground truth start, ground truth future, and predictions for the robot-aware model and VFS model. The baseline blurs the bear in the top row, and predicts minimal movement for the octopus in the bottom row. The proxy robot is visualized in green. Refer to the videos on the website for more visual comparisons.

centimeters. Similar to RoboNet, the states are normalized by the workspace minimum and maximum. We also collected 25 trajectories of the modified WidowX200 as seen in 4 for evaluation.

### A.3 SIMULATED CONTROL EXPERIMENT DETAILS.

We study the effect of robot transfer on pushing and pick-and-place tasks using the MuJoCo simulation (Todorov et al., 2012). The models are trained on 10k videos of the WidowX200 and evaluated on their task performance on the unseen Fetch robot.

**Pushing.** The robot must push the block from a randomized starting pose to a randomized goal position 10cm away from the start. To initialize the environment, we first set the robot and block spawn locations. The block is initialized with uniform noise of 5cm in the center of the workspace. The robot gripper is then moved behind the block, and then perturbed with uniform noise of 1cm. The episode limit is 10 steps. We only used the image-based world cost for pushing so we set $\lambda = 0$.

**Pick-and-place.** The robot must pick an object from a randomized starting pose and place it on an elevated platform at a location specified through a goal image. For pick-and-place, we give a goal for picking, lifting, and placing for a total of 3 goals per pick-and-place episode. We found it helpful to use the robot cost term in Equation 3, specifically by using the $L_2$ distance between the current and goal end effector position. We scaled $\lambda$ so the magnitude of world and robot cost would roughly contribute equal weight to the total cost. Note that using only the robot cost term causes the agent to completely focus on arm placement and ignore block placement. To initialize the environment, we first set the robot and block spawn locations. The block is initialized with uniform noise of 6cm next to the platform. The robot gripper is then moved above the block, and then perturbed with uniform

noise of 3cm. We give the controller up to 5 steps to achieve the current goal, before switching to the next goal for a maximum episode length of 15 steps.

## A.4 CYCLEGAN BASELINE.

To the best of our knowledge, no prior methods can perform zero-shot transfer of visuomotor skills between robots. However, we have looked into alternative approaches that might plausibly perform few-shot transfer between robots. In particular, based on the human-to-robot transfer approach of Smith et al. (2020), we can train a CycleGAN to translate images of the test-time robot to the train-time robot. Then, having trained full robot+world dynamics models on the train-time robot, we may apply them directly to the translated images for accomplishing robotic tasks. This is also similar to transfer approaches proposed in Rao et al. (2020); Raychaudhuri et al. (2021) and Roy & Konidaris (2021).

We used the official CycleGAN implementation with default hyperparameters. The training process used 1k videos (12k images) per robot. We trained the CycleGAN until convergence, and the CycleGAN outputs look visually accurate (see website).

## A.5 REAL WORLD CONTROL EXPERIMENT DETAILS.

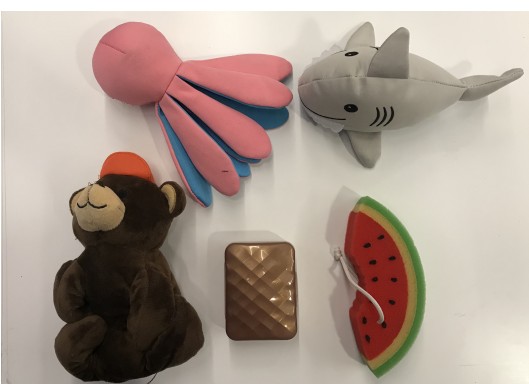

Figure 8: The objects used in the control experiments. The few-shot WidowX250 transfer experiment used all objects, while the zero-shot Franka experiment used the bear and watermelon.

The bear, watermelon, box, octopus, and shark used in the control experiments are seen in Figure 8, and vary in size, texture, color, and deformability.

In the few-shot WidowX200 control experiment, the robot is tasked with pushes on five different objects that vary in shape, color, deformability, and size. We chose two directions, a forward and sideways direction, giving a total of 10 push tasks. We run 5 trials for each push task for a total of 50 pushes per method. For the zero-shot Franka experiment, the controllers are evaluated on two different objects pushed in three different directions. Each pushing task is repeated 5 times for a total of 30 trials per controller.

For the cost function, we only used the image-based world cost function $C_w$ by setting $\lambda = 0$.

**CEM action selection.** As mentioned in Section 3, the CEM algorithm is used to search for action trajectories that minimize the given cost function. The CEM hyperparameters are constant across controllers to ensure that all methods get the same search budget for action selection. For the few-shot WidowX200 experiment, the CEM action selection generates 300 action trajectories of length 5, selects the top 10 sequences, and optimizes the distribution for 10 iterations. For the zero-shot Franka experiment, the CEM action selection generates 100 action trajectories of length 5, selects the top 10 sequences, and optimizes the distribution for 3 iterations.

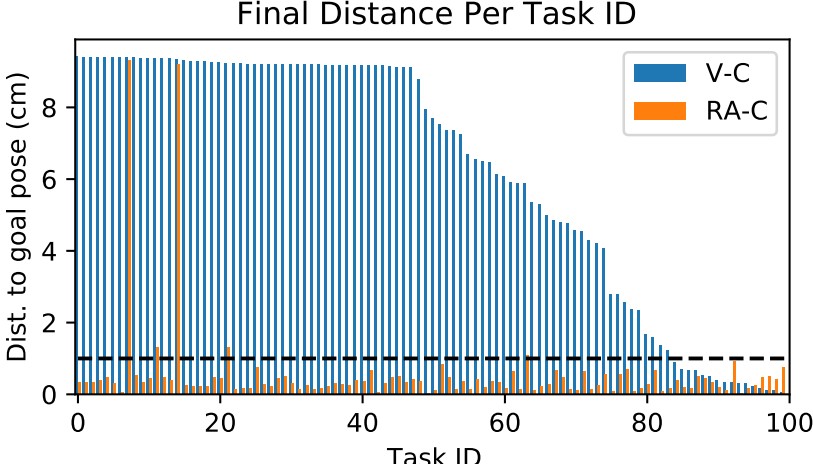

Figure 9: We evaluate the performance of the pixel cost (V-C) and robot-aware cost (RA-C) by running 100 push tasks where the robot must achieve the object-only goal image. Then we visualize for each task, the final pose distance between the object and the object pose in the goal image. The dashed horizontal line indicates the success threshold of 1cm. The robot-aware cost succeeds in most tasks by outputting a sensible cost over the world region, while pixel cost is distorted by the robot.

### A.6    EVALUATING THE ROBOT-AWARE COST IN ISOLATION

The control experiments in performed in Section 4 evaluate the robot-aware cost with various choices of dynamics models. Here, we evaluate the cost function more closely in isolation by using ground-truth dynamics of the simulator instead of a learned model.

In section 3.1.2, we propose to separate the conventional pixel-wise cost into a robot-specific and world-specific cost. We report the results of a simulated experiment, where we set up a block pushing environment with the Fetch robot. The environment consists of three objects with varying shapes, colors, and physics. We sample the goal image by moving one of the objects from its initial pose to a random pose 10cm away.

We then run the visual foresight pipeline using *ground truth* dynamics and the given cost function for 5 action steps, and record the final distance between the object and its pose in the target image. Success is defined as moving the object within 1cm of the goal pose. As seen in Figure 9, The robot-aware cost (RA-C) gets 95% success rate, where as pixel cost (V-C) only gets 16% success rate.

The robot-aware cost is able to disregard the extraneous robot when computing the pixel difference between the current image with the robot and the goal image without the robot. Pixel cost on the other hand, results in the CEM selecting actions that move the robot out of the scene rather than moving the object to the correct pose.

### A.7    IMITATING HUMAN GOAL IMAGES.

We evaluate the robot-aware cost's effectiveness on achieving goal images with a human arm. Videos are on the project website. We collect five goal images by recording human pushing demonstrations, and use the last image from each video as the goal image as seen in Figure 10. Human masks are annotated using Label Studio (Tkachenko et al., 2020-2021). We run RA/RA and RA/Pixel controllers, which differ only in the planning cost. RA/RA achieves all goal images, whereas RA/Pixel fails on all goal images. See the website for videos.

### A.8    USING ADDITIONAL CAMERA VIEWS

Next, we analyze the benefit of using additional viewpoints for the robot-aware cost function. One potential local minima of the robot-aware cost is occlusion of the object by the gripper, since robot-

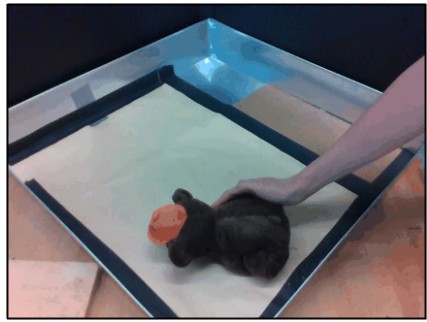

Figure 10: Example human goal image.

aware cost does not compute costs over the robot region. However, we did not find this occlusion behavior to be a problem in practice since our CEM search was powerful enough to find the true solution (which has lower cost than the occlusion minima). We can also address this by adding an additional viewpoint. We tested the multi-view version of our cost function on the simulated pushing setup, and found it improved performance as seen in Table 4.

Using multiple views gave larger performance gains in the pick-and-place domain. Here, we repeat the pick-and-place experiment from section 4 by training a multi-view video prediction model on the WidowX, and then transfer it to a multi-view scene of the Fetch robot.

Table 4: Adding an additional view improves pick-and-place performance.

| Dynamics Model | Cost | Fetch Pick-and-Place (1 View) | Fetch Pick-and-place (2 Views) |
|---|---|---|---|
| VF+State | Pixel | 0/20 (0%) | 0/20 (0%) |
| RA | Pixel | 0/20 (0%) | 0/20 (0%) |
| VF+State | RA | 4/20 (20%) | 8/20 (40%) |
| RA | RA | **8/20** (40%) | **14/20** (70%) |

### A.9 EFFECT OF ROBOT APPEARANCE ON PREDICTION

In addition to the real-world transfer examples reported in the main paper, we experiment with robot transfer in MuJoCo simulations (Todorov et al., 2012) by setting up a tabletop manipulation workspace. During training, the model is trained on 10,000 videos of a gray WidowX200 5-DOF robot performing random exploration. Then, we evaluate the model on various modifications of the WidowX. First, we try changing the color of the entire WidowX from gray to red. Next, we extend the forearm link of the WidowX by 10cm, similar to the real-world modified WidowX experiment in Section 4. Finally, we evaluate on a WidowX with a longer and red link. As seen in Table 5, the robot-aware model outperforms the vanilla model in world PSNR and SSIM metrics.

For the color change, the vanilla model predicts a still image for all timesteps, which suggests that the network does not recognize the red robot as the training time robot. Our model is invariant to the color shift of the robot due to the masking, and can accurately predict the trajectory with little degradation in quality.

With the link length change, the vanilla model is able to predict robot movement, but it replaces the longer link with the original short link. In some cases, the longer link allows the robot to contact the object and move it. Our model correctly predicts the object movement, but the vanilla model fails to model the object interaction and movement since the object contact is not possible with the shorter link.

Finally, in the longer and different color link setting, the vanilla model is able to recognize and predict movement for the unaltered parts of the robot. It replaces the long red link with the original short gray link, and leaves the long red link in the image as an artifact. Similar to the previous experiment, the vanilla model has degraded object prediction while our model can still accurately predict the dynamics.

Table 5: Zero-shot video prediction results to a modified WidowX200 in simulation. The modified WidowX200 has different color and/or link lengths.

| | Color Change | | Link Change | | Color and Link Change | |
|---|---|---|---|---|---|---|
| | PSNR ↑ | SSIM ↑ | PSNR ↑ | SSIM ↑ | PSNR ↑ | SSIM ↑ |
| VF | 37.5 | 0.976 | 42.3 | 0.994 | 40.3 | 0.987 |
| RA (Ours) | **38.7** | **0.985** | **45.2** | **0.997** | **40.9** | **0.991** |

### A.10   RAC PERFORMANCE ON TRAINING ROBOT

As a sanity check, we tested if the baseline and our method is suitable for planning on the original robot. We follow the exact setup as the simulated pushing and pick-and-place experiments, except that we do not change the robot during evaluation. As seen in Table 6, our method is marginally better in performance in prediction and control in comparison with the baselines.

Table 6: Performance on the training robot.

| Dynamics Model | Cost | Push Success | Pick-and-place Success |
|---|---|---|---|
| VF+State | Pixel | 86/100 (86%) | 9/20 (45%) |
| RA | RA | **92/100** (92%) | **11/20 (55%)** |

### A.11   ROBOT MASKS AND CALIBRATION.

Acquiring robot masks that accurately segment the robot and world is crucial for the robot-aware method, as it depends on the mask to train the dynamics model and evaluate costs as mentioned in Section 3.1.1 and 3.1.2. The first step to acquiring robot masks is to get the camera calibration, which is the intrinsics and extrinsics matrix of the camera. If we have physical access to the robot and camera, acquiring the camera calibration is trivial. In our experiment setup with the Franka and WidowX200, we use AprilTag to calibrate the camera extrinsics, i.e. the transformation between camera coordinates and robot coordinates given the camera intrinsics.

**Extracting calibration from RoboNet.**   However, if we do not have access to the robot and camera, as in RoboNet, acquiring camera calibration is still possible. RoboNet does not contain the camera extrinsics information, but does contain the camera model information. Therefore, we use the default factory-calibrated camera intrinsics for each corresponding camera model.

Next, RoboNet contains the 3D positions of the robot end-effector in the robot coordinates for each image. For each viewpoint, we hand-annotate a few-dozen 2D image coordinates of the corresponding end-effector, and use OpenCV's camera calibration functionality to regress the camera extrinsics given the camera intrinsics and labeled 3D-2D end-effector point pairs.

**Synthesizing masks.** Once we have the camera calibration of the robot, we render the 3D model of the robot, and project it to a 2D segmentation map using the camera calibration. We use the MuJoCo simulator for this process. Conveniently, the MuJoCo simulator can render a segmentation map of the geometries for a given camera viewpoint. By setting up an empty MuJoCo scene with only the robot geometries, we can render the geometry segmentation map and use that as the robot mask.

**Using depth.**    Observe that the robot mask $M_r(t)$ above is computed from only the robot state, without any reference to the world state. As a result, it can not account for robot occlusion by objects in the scene. For example, if the robot is pushing a large object towards the camera, the part of the object occluding the robot will count as the robot region. With RGB-D observations $o(t)$, which are commonly used in robotics, it is easy to refine $M_r(t)$ to remove occluded regions. To do so, we may compute the distance $D_r(x,y)$ of all robot pixels through the above projection, and zero out pixels in the mask where $D_r(x,y)$ is greater than the observed depth at that pixel. In other words, pixels that are closer to the camera than the computed distance of the robot at that location must correspond to occluders. In practice, in our experiments, we find it sufficient to use RGB cameras and ignore occlusions.

