# OpenReview forum: "Know Thyself: Transferable Visual Control Policies Through Robot-Awareness"
_ICLR.cc/2022/Conference — ICLR 2022 Poster_

### Official Review · Reviewer_rNxF · 2021-11-02

**Correctness:** 3
**Technical Novelty And Significance:** 3
**Empirical Novelty And Significance:** 3
**Recommendation:** 8
**Confidence:** 4

**Main Review:**

 Strengths:
* A nice example of a method that cleverly utilises "knowledge we already know" rather than trying to learn it from data!
* The overall idea is well-explained with a sufficient number of illustrative figures and experiments, both in simulation and in the real world, across multiple platforms and datasets (e.g. RoboNet).
* Very interesting few-shot and zero-shot transfer results!
* Thorough analysis has been performed on the experiments. Both quantitative and qualitative results are presented.

Weaknesses and Suggestions:
* The method used to project and segment a robot shape into the image plane of a camera is a central component that makes RAC possible but not much is said about it.
* _Disentangled_ might not be the best term here - in the representation learning literature we could claim two things are disentangled from each other if they are independent and contain little mutual information for each other - e.g. dimensions in a multi-dimensional learned feature space of a GAN, VAE. In the presented method we do separate robot pixels from world pixels when deciding what will P_w see during its training. However, that does not mean that P_w does not contain information about the robot's movement too - it operates over the masks of the robot in the images, which evolve over time. If that is not the case it should be better explained exactly what image information does P_w see and how is the robot information factored out such that P_w is disentangled from the robot's dynamics.
* The notion of a Proxy Robot is mentioned in Section 3.2 - this was a bit confusing. Aren't you just masking out the robot pixels or _"...projecting a black "proxy” robot over the true robot in all images..."_ refers to something else?
* Given that the evaluation is over multiple videos, you could provide standard deviation statistics for the results in Table 1, together with the mean estimates that are already there (if it would not clutter the table too much)
* Choosing goal images that only have objects in them and no robot arms, in the experiments for transferring visual policies across robots, might be putting the baselines at an unfair disadvantage - why not 50/50 - half the images have robot arms and the other half don't?
* A few words on the the limitations of the method would be very useful - e.g. how much is RAC dependent on proper calibration - both for the camera'a parameters and the relative pose of the camera with respect to the observed robot. How good are camera-to-robot calibration methods for robots that look similar - e.g. a UR3 vs UR5 vs UR10?
* It's rarely the case that the world stay the same and only the robot would change - objects could change colour and size, camera pose might change, scene illumination might change, etc. Being robust to such perturbations is no less important. However, given the robot-aware nature of the work, keeping the world fixed is a fair assumption for this paper's contributions.

**Summary Of The Paper:**

The presented work is can be seen as a more sample efficient improvement/augmentation of the Visual Foresight method (Ebert et al.) in the context of transfer learning for robots. The authors present the notion of robot-aware models - the overall dynamics model is explicitly decomposed onto a known analytical dynamics model for the robot and a learned dynamics model (from pixels) for the objects in the scene. The analytical model, combined with off-the-shelf image plane projection methods, tell us how a robot would look like from the camera's PoV after a set of taken actions. This in turn allows us to exclude any robot-related pixels from the scene observations and provide the learned dynamical model only with object-related pixel information, effectively making the learned model invariant to the robot's appearance. The same logic is applied to a novel planning cost function used for performing visual MPC. As a result cross-robot transfer can be more easily facilitated in the context of table-top manipulation tasks, where the goal is specified through a goal image.

**Summary Of The Review:**

Overall, this paper does present a novel solution to the problem of robust transfer learning on robots in a sample efficient fashion. It is a good example of how to successfully leverage information we have about the problem a priori (robot dynamics) but still learn anything else we don't know (object dynamics) from data. This is proposed as an alternative to the otherwise brittle methods which try to learn everything from data, end-to-end. While there are minor details to be elaborated on and fixed, the idea itself and the evidence provided here are sufficient for publication and are a useful contribution to the field.

---

### Official Review · Reviewer_U6rV · 2021-11-03

**Correctness:** 3
**Technical Novelty And Significance:** 4
**Empirical Novelty And Significance:** 3
**Recommendation:** 6
**Confidence:** 3

**Main Review:**

The method presented in this paper is interesting and seems to fill a gap in the existing literature.

The paper is written clearly and presents the concepts in a cohesive way. The experimentation is fairly convincing and uses both simulated and real data.

The papers suffers from the following limitations (sorted by decreasing order of severity):

The paper starts and ends with a bold claim about the fact that it permits "zero-shot transfer onto new robots for the first time".  This is not the case, as the literature review itself shows.  That being said, the paper does seem to fill a gap in the literature.  It would make sense to re-calibrate this claim.

The paper does not provide enough technical details about the models' implementation.  It is therefore hard to conclude whether the approach is technically sound. Section 3.2 that provides implementation details falls short of providing enough information.

The complementarity to relevant work in Section 5 should be discussed in greater details.  In particular, the comparison with Chen et al. (2018) should be clarified.

In Section 2, the paper mentions that "When parts of this projection may be occluded, we may further use RGB-D observations to easily identify and handle them". This approach is debatable, as the appearance of the robot and of the world could be very similar on this metric in the real-world.

A number of references must be corrected (for example, reference 7 seems to be an ICML paper but mentions PMLR, reference 2 lacks details, etc.)

**Summary Of The Paper:**

This paper proposes a robot-aware RL-based approach to transfer learning from a robot to another. The approach factorizes the model into two models, a robot model and a world model. The method is evaluated against several baselines on simulated and real data. The paper claims that it permits zero-shot transfer onto new robots for the very first time.

**Summary Of The Review:**

This paper provides an interesting approach to zero-shot learning but falls short of providing enough technical details about the implementation. The complementarity to existing work is not discussed enough.  For these reasons, I advocate for a reject.

EDIT AFTER DISCUSSION PERIOD: The authors have addressed the majority of my concerns.  Reading their answers and the discussions with other reviewers, I updated several scores upward and my final score to "marginally above".

---

### Official Review · Reviewer_BJUg · 2021-11-04

**Correctness:** 4
**Technical Novelty And Significance:** 3
**Empirical Novelty And Significance:** 3
**Recommendation:** 6
**Confidence:** 4

**Main Review:**

The paper works on a quite interesting problem and provides an intuitive approach. The neural network architectue is based on an extension of stochastic video generation architecture using LSTM. The overall design of the system as described in the main text looks interesting and seems to perform well on certain tasks. Overall, it is a bit hard to follow the details of the methodology in the main text, especially in terms of presenting the novel contributions in this paper. The algorithm for testing in Appendix definitely is helpful, is it possible to add one for the training phase to summarize? It would be interesting to see a discussion on why the pick-and-place results are always much lower than pushing results. Real robot experiments seems to be limited, 50 pushesfor few-shot experiments, 30 for zero-shot.
Minor: Could you clarify what was the final form of Eq 3, how was it used in the final system? How was Lambda chosen? In experiments section, there is a repetition: "1.8k videos videos", so please do a thorough proofread.

**Summary Of The Paper:**

This paper addresses the problem of how to leverage data collected from on a robot to reduce/remove the need for robot specific data. The authors propose a model based RL policy based on coupling robot-agnostic and robot-specific dynamcis modules. Experiments with simulated and real robots are demonstrated, showing transferability of visual model based policies.

**Summary Of The Review:**

The paper addresses an interesting problem, presents good results, needs some clarifications.

---

### Official Review · Reviewer_2sNR · 2021-11-07

**Correctness:** 3
**Technical Novelty And Significance:** 2
**Empirical Novelty And Significance:** 2
**Recommendation:** 5
**Confidence:** 3

**Main Review:**

The paper is an attempt to enhance previous works on visual foresight by adding the information coming from the analytical model of the robot forward kinematics. The basic idea is to use such model to analytically compute a mask to decouple the robot specific visual appearance from the task-relevant world appearance. Decoupling world-specific and robot-specific visual information attempts at avoiding a limitation of non-decoupled visual foresight, which doesn't allow to privilege world specific visuals with respect to robot-specific visuals.

The paper is well written and easy to read even though some important details (e.g. the definition of the proxy-robot) require the reader to read carefully the supplementary material.

An important limitation of the paper is that authors fail at explaining if in the world observation the robot information is completely removed from the information related to objects present in the scene (which is what I was expecting as a natural improvement from the original VF paper). Even after reading the paper a few times, it actually seems that some robot information is retained in the world observation (i.e. as mentioned in A.1 "Proxy-robot. An image o is composed of the robot image or = o ◦ Mr and world image ow = o ◦ (1 − Mr) where Mr is the mask of the robot. We propose to edit all images ow = ProxyRobot(o, M)".). If that's the case, it seems that even ignoring completely the robot aware cost in Eq. 3 (by setting λ to zero) wouldn't remove completely the robot-specific information.

A minor limitation of the paper is that it is incremental leveraging a significant amount of prior works (e.g. Smith at al. 2020) without being clear about how prior work has been accessed to (e.g. authors' implementation, open-source release or other).

= SOME OTHER COMMENTS =
- Page 6. "The current and future mask are concatenated with the image, and the end effector pose is tiled into the latent spatial embedding".  This sentence isn't clear. Which image and which latent space are authors referring to?

- PSNR and SSIM haven't been defined and it would be better if authors could define them in the paper.

**Summary Of The Paper:**

The paper presents an extension of previous works on training a visual policy to control a robot arm in a manipulation context. The policy is pre-trained on a large robot experience dataset (e.g. different robots interacting with different objects in a table-top setting). The dataset is used to train a model for the visual dynamics (i.e. next visual & robot state given current visual & robot state and applied action). The visual dynamics are used to reach a desired visual state using an existing technique: visual foresight (VF) by Finn & Levine, 2017 and Ebert et al., 2018.

The main novelties of the paper of the paper are two. First, authors propose a technique to learn a decoupled visual dynamic model: the world model (i.e. a model to propagate forward the world-observations consisting of an image of all objects in the scene excluding the robot) is decoupled from the robot model (i.e. a model to propagate forward the end-effector pose and a mask which represents the robot body in the visual space). Second, the visual foresight planning relies on a newly proposed cost which combines and decouples two costs: a desired robot-state cost and a desired world-state cost.

**Summary Of The Review:**

As mentioned above, the major limitation of the paper is its limitation in explaining to which extent the world model is affected by the robot model and by the state of objects in the scene. Ideally, a natural evolution of previous approaches would be to provide a world model completely decoupled from the robot model and mainly influenced by the state of objects in the scene.

Even after reading the paper carefully, it's still unclear to me if the world model proposed by the authors is mainly used to propagate forward the information of the state of objects in the scene. Considering that the authors rely on an analytical model of the robot, this is an important requirement and I would like to be convinced that either this is already a property of the proposed world model or that this is not the case for good/understandable/sound reasons.

---

### Decision · Program_Chairs · 2022-01-20

**Decision:**

Accept (Poster)

**Comment:**

This paper proposes a method to improve the transfer of visual control policies between robots.  The method extends a visual foresight approach using a learned robot-agnostic world-dynamics model and a (potentially analytic) robot-specific robot-dynamics module.  A key aspect of the method is to form a blocky mask over the robot's body in the visual image, thus allowing the learned dynamics to depend less on appearance attributes of the robot.  Planning with these dynamics models operates in the visual observation space.  The method is tested for zero-shot transfer on multiple physical robots and also with simulated robots, with positive results across multiple experiments.

The reviews raised multiple concerns on the details of the method and the clarity of the presentation that were largely addressed in the author response.  The authors refined their claims to a demonstration of zero-shot transfer of visual skills across real-world robots.  The evaluation of the proposed transfer method on real-world robots is a notable strength of the paper.  The core limitation, raised by one reviewer, is that the generality of the invariance is only provided by a visual mask on the color and appearance of the robot.  This serves as a limited form of invariance to the robot's dynamics.

The discussion phase did not yield a consensus on the merits of the paper, with the proposed method seen as useful but still limited.  Three knowledgeable reviewers indicate to accept the paper and one indicates to reject.  The formal description does not make the world-dynamics explicitly robot-agnostic, as the learned model $(P_w(o_w'|o_w, r, r', a)$ in Equation 2) still explicitly depends on the actions of the robot ($a$) and thus the robot's dynamics.  Despite some limitations in the formalism, the practical utility of the method is convincingly demonstrated in multiple robot experiments.  This is a clear contribution to the literature, which is supported by three reviewers.  The paper is therefore accepted.